# Tryptase in Acute Appendicitis: Unveiling Allergic Connections through Compelling Evidence

**DOI:** 10.3390/ijms25031645

**Published:** 2024-01-29

**Authors:** Nuno Carvalho, Elisabete Carolino, Margarida Ferreira, Hélder Coelho, Catarina Rolo Santos, Ana Lúcia Barreira, Susana Henriques, Carlos Cardoso, Luís Moita, Paulo Matos Costa

**Affiliations:** 1Serviço Cirurgia Geral, Hospital Garcia de Orta, 2805-267 Almada, Portugal; margarida.s.f@gmail.com (M.F.); analuciapb19@gmail.com (A.L.B.); susanahenriques@campus.ul.pt (S.H.); paulomatoscosta@gmail.com (P.M.C.); 2Faculdade Medicina, Universidade Lisboa, 1649-028 Lisboa, Portugal; 3H & TRC—Health & Technology Research Centre, ESTeSL—Escola Superior de Tecnologia da Saúde, Instituto Politécnico de Lisboa, 1549-020 Lisboa, Portugal; etcarolino@estesl.ipl.pt; 4Serviço de Anatomia Patológica, Hospital Garcia de Orta, 2805-267 Almada, Portugal; hmoc85@gmail.com; 5Serviço de Cirurgia Geral, Hospital de Nossa Senhora do Rosário, 2830-003 Barreiro, Portugal; catarinarolo9@gmail.com; 6Dr. Joaquim Chaves Laboratório de Análises Clínicas, 1495-068 Algés, Portugal; carlos.cardoso@jcs.pt; 7Innate Immunity and Inflammation Lab., Instituto Gulbenkian de Ciência Oeiras, 2780-156 Oeiras, Portugal; lmoita@igc.gulbenkian.pt; 8Instituto de Histologia e Biologia do Desenvolvimento, Faculdade Medicina, Universidade Lisboa, 1649-028 Lisboa, Portugal

**Keywords:** acute appendicitis, allergy, histamine, hypersensitivity type I reaction, mastocytes, pathophysiology, serotonin, tryptase

## Abstract

The aetiology of acute appendicitis (AA), the most frequent abdominal surgical emergency, is still unclarified. Recent epidemiologic, clinical and laboratorial data point to an allergic component in the pathophysiology of AA. Mastocytes participate in the Th2 immune response, releasing inflammatory mediators from their granules upon stimulation by IgE-specific antigens. Among the well-known mediators are histamine, serotonin and tryptase, which are responsible for the clinical manifestations of allergies. We conducted a prospective single-centre study to measure histamine and serotonin (commercial ELISA kit) and tryptase (ImmunoCAP System) concentrations in appendicular lavage fluid (ALF) and serum. Consecutive patients presenting to the emergency department with a clinical diagnosis of AA were enrolled: 22 patients with phlegmonous AA and 24 with gangrenous AA The control group was composed of 14 patients referred for colectomy for colon malignancy. Appendectomy was performed during colectomy. Tryptase levels were strikingly different between histological groups, both in ALF and serum (*p* < 0.001); ALF levels were higher than serum levels. Tryptase concentrations in ALF were 109 times higher in phlegmonous AA (APA) (796.8 (194.1–980.5) pg/mL) and 114 times higher in gangrenous AA (AGA) (837.4 (272.6–1075.1) pg/mL) than in the control group (7.3 (4.5–10.3) pg/mL. For the diagnosis of AA, the discriminative power of serum tryptase concentration was good (AUC = 0.825), but discriminative power was weak (AUC = 0.559) for the differential diagnosis between APA and AGA. Mastocytes are involved in AA during clinical presentations of both phlegmonous and gangrenous appendicitis, and no significant differences in concentration were found. No differences were found in serum and ALF concentrations of histamine and serotonin between histological groups. Due to their short half-lives, these might have elapsed by the time the samples were collected. In future research, these determinations should be made immediately after appendectomy. Our findings confirm the hypersensitivity type I reaction as an event occurring in the pathogenesis of AA: tryptase levels in ALF and serum were higher among patients with AA when compared to the control group, which is in line with a Th2 immune response and supports the concept of the presence of an allergic reaction in the pathogenesis of acute appendicitis. Our results, if confirmed, may have clinical implications for the treatment of AA.

## 1. Introduction

Acute appendicitis (AA) is a major cause of acute abdominal pain in adults and one of the most common reasons for emergency surgery [1]. The lifetime incidence risk in the general population is 6.7% to 8.6% [2].

Based on histological features, namely eosinophilic infiltration, MC degranulation, and muscular oedema, an allergic component was proposed in the aetiology of AA [3].

Recent evidence suggest that the hypersensitivity type I reaction is an event in AA [4,5,6,7]. In the presence of a specific antigen, Th2 cells secrete IL-4, 5, 9 and 13. IL-4 induces B cell production of IgE, which in turn acts on specific receptors in MCs and triggers histamine, serotonin and tryptase release [8,9,10]. IL-5 is the primary cytokine involved in the recruitment and activation of eosinophils [11], and IL-4 and IL-9 induce the growth of mast cells and basophils [9].

Cytokines associated with the Th2 response are present in the ALF of patients with AA [5]. Children with IgE-mediated allergies have an adjusted risk of developing complicated appendicitis that is three times lower than that of those without IgE-mediated allergies [4]. A study showed significantly increased numbers of IgE-positive cells in phlegmonous AA when compared to that in incidental appendectomy [6]. Cationic eosinophilic proteins are elevated in serum and ALF in patients with AA [7].

Mast cells (MCs) play a central role in IgE-mediated allergic responses [12,13]. MC activation occurs when the IgE antibody binds to the respective antigen, which in turn causes crosslinking of the specific receptor on the MC surface and triggers degranulation, releasing pre-packaged cytoplasmic granules that contain a variety of biologically active substances including histamine, serotonin and several proteases like tryptase [8,10,13,14].

MCs are key components of the host immune defence and are strategically located at host-environment interfaces, like the intestinal mucosa, where they may act as sentinels that sense pathogens and initiate metabolic immune responses [12,13,14].

However, the pro-inflammatory and cytotoxic substances released by activated MCs may result in damage to normal tissues [13], turning their protective interaction into one that is pathogenic.

Experimental models using KIT-dependent and KIT-independent mast cell-deficient mice and mice with specific mast cell-mediator deficiencies suggested that MCs can either promote host resistance to infection or contribute to a dysregulated immune response that can increase host morbidity and mortality [14].

MC degranulation is a hallmark of allergy [15]. Serotonin and tryptase are involved in allergies and histamine is the final effector of allergic reactions—responsible for its pathogenic and clinical manifestations [16,17]. Histamine induces smooth muscle contraction, vasodilation, increased vascular permeability and chemotaxis for neutrophils and eosinophils [18,19].

Serotonin induces smooth muscle contraction and mucus production [20]. The function of tryptase has not been completely elucidated; it has numerous proinflammatory functions and serum concentrations are elevated in several pathological conditions involving allergic reactions [21].

Toluidine blue is used for staining mast cells and can be an indirect method of measuring MC degranulation [22]. In a previous study, MC counts per cross-sectional area were evaluated in AA and normal appendices [22]. The number of MCs stained was lower in AA. This proved that MC degranulation is increased in AA compared with normal appendices [22].

If an allergic component were to be present in AA, it would be expected that histamine, serotonin and tryptase levels would be higher in AA when compared to the control group. We previously developed the concept of appendicular lavage fluid (ALF) for investigating local immunoinflammatory changes in AA [5,7,11]. ALF is a unique procedure, which, unlike other methods, allows the local harvesting of fluid with cells and inflammatory markers.

We evaluated the levels of histamine, serotonin and tryptase in the ALF and peripheral blood of patients referred for appendectomy due to a clinical diagnosis of AA.

Basophils have a common origin and share several biochemical and functional features with MCs [23,24,25]. We also evaluated the serum levels of basophils in the context of AA.

### Objective

This study aimed to evaluate the contribution of mast cells in acute appendicitis by measuring histamine, serotonin and tryptase levels in the appendicular lavage fluid and peripheral blood of patients with a clinical diagnosis of acute appendicitis. As basophils share several characteristics with mastocytes and are easily available in peripheral blood, our secondary objective was to determine their peripheral blood counts. The data obtained were used to further test our hypothesis that an allergic reaction is implicated in the pathogenesis of AA.

## 2. Results

### 2.1. Demographic Data

We analysed a total of 46 patients with a clinical diagnosis of AA; they were categorized according to histological features: 22 patients had APA and 24 patients had AGA. As a control group (*n* = 14), patients submitted for right colectomy for cancer were included (Table 1), all of whom had normal appendicular histology. The median age of patients with AA was 44 (29–60) years old.

Significant differences among control, APA and AGA groups were found regarding age (*p* = 0.044). After performing pair-wise analysis, this difference was shown to have been maintained in the pair Control–AGA (*p* = 0.021), with the control group being the oldest of them all.

No differences were found related to gender, presence of allergy, BMI and time elapsed between the first clinical manifestations of AA and appendectomy (Table 1). BMI was higher in patients with AGA (Table 1). Four patients had a history of allergies, but none took anti-allergy medication in a regular fashion.

### 2.2. Hemogram and Appendicular Histology

Hemogram and appendicular histology data are depicted in Table 2.

WBC count was different between the control, APA and AGA groups (*p* < 0.001) and this difference was significant in the pair APA–Control (*p* < 0.001) and in the pair AGA–Control (*p* < 0.001). The control group had the lowest levels.

Concerning neutrophils, significant differences were also found (*p* = 0.005) and maintained in the pair AGA–Control (*p* = 0.004), with the highest serum levels found in AGA cases.

No differences were found concerning serum concentrations of lymphocytes.

Regarding eosinophils and basophils, marginal significant differences were found. Higher levels of eosinophils and lower levels of basophils were found in patients with APA (Table 2).

Regarding monocytes, the differences found (*p* = 0.034) between groups and maintained in the pair AGA–Control (*p* = 0.028) were due to levels in the control group being higher.

### 2.3. Histamine, Serotonin and Tryptase Serum Levels

Serum histamine, serotonin and tryptase levels are presented in Table 3 and Figure 1a–c.

Mastocyte granule proteins were detected and measured in all the patients. The concentrations of histamine (*p* = 0.948) and serotonin (*p* = 0.290) were not significantly different between the histologic groups (Table 3) (Figure 1a–c).

Tryptase levels were found to be significantly different amongst the groups (*p* = 0.001) and these differences were maintained in both the pairs AGA–Control (*p* = 0.014) and APA–Control (*p* = 0.001), with the lowest levels having been found in the control group (Figure 1).

### 2.4. Appendicular Lavage Fluid Histamine, Serotonin and Tryptase Levels

Appendicular lavage fluid histamine, serotonin and tryptase levels are presented in Table 4 and Figure 2a–c.

Mastocyte granule proteins were detected in the ALF of all the studied subjects.

Levels of histamine (*p* = 0.756) and serotonin (*p* = 0.474) were not different between the histologic groups (Figure 2a,b).

Concerning tryptase levels, significant statistical differences were found (*p* = 0.001) and maintained for the pairs AGA–Control (*p* = 0.004) and APA–Control (*p* = 0.001), with the highest levels being related to the AGA and APA groups. No differences were found between the AGA–APA pair (Figure 2c).

Tryptase serum levels and AA diagnosis were evaluated by ROC curve analysis. (Figure 3). We evaluated the discriminatory power of tryptase serum levels for distinguishing APA from AGA. The area under the curve (AUC) for tryptase, the optimal sum of sensitivity (0.731) and specificity (0.550) at a cut-off level of greater than 50.75 ng/mL, was 0.559 (*p* = 0.505).

### 2.5. Relationship between Serum Histamine, Serotonin and Tryptase Levels and Appendicular Lavage Fluid (ALF) Histamine, Serotonin and Tryptase Levels

The relationship between MC granule protein concentrations in the serum and in appendicular lavage fluid is depicted in Table 5.

Several significant correlations were found. Serum histamine concentrations correlated with ALF histamine concentrations (*p* < 0.001). Serum tryptase concentrations correlated with both ALF tryptase concentrations (*p* < 0.001) and serum histamine concentrations (*p* = 0.001).

The ALF levels of histamine and tryptase were correlated (*p* = 0.001).

Serum serotonin concentrations correlated with serotonin concentrations in ALF (*p* < 0.001).

### 2.6. Relationship between Blood Basophils and Serum Histamine, Serotonin and Tryptase Levels, and Appendicular Lavage Fluid Histamine, Serotonin and Tryptase Levels

The relationship between peripheral blood basophil count (PBBC) and MC granule protein concentrations in serum and appendicular lavage fluid is portrayed in Table 6.

A positive correlation was found between PBBC and histamine concentrations in both peripheral blood (*p* = 0.002) and ALF (*p* = 0.019). A negative correlation was found between BPBC and serotonin concentrations in ALF. No correlation was found between PBBC and tryptase.

### 2.7. Relationship between Serum Histamine, Serotonin and Tryptase concentrations and Acute Appendicitis Clinical Presentations

Table 7 depicts the relationship between serum histamine, serotonin and tryptase concentrations, appendicular perforation (yes or no) and the presence (yes) or absence (no) of peritonitis.

Patients with peritonitis had higher levels of serum tryptase (*p* = 0.020).

The area under the curve (AUC) for serum tryptase concentration in the diagnosis of peritonitis, which has an optimal sum of sensitivity (0.750) and specificity (0.375) at a cut-off level of values greater than 56.95 ng/mL, was weak (0.685).

### 2.8. Relationship between ALF and Histamine, Serotonin and Tryptase Concentrations and Acute Appendicitis Clinical Presentations

The relationship between ALF histamine, serotonin and tryptase levels, appendicular perforation (yes or no) and the presence (yes) or absence (no) of peritonitis is depicted in Table 8.

Significant differences were only found for tryptase levels and peritonitis, with higher levels of tryptase having been detected whilst in the presence of peritonitis (*p* = 0.030).

## 3. Discussion

We evaluated MC granule inflammatory mediators in the ALF and blood of patients with AA. Higher levels of tryptase were found in AA in both the ALF and blood compared with in the control group.

Although the aetiology of AA remains unclear [26], luminal obstruction is thought to be one of the possible causes [27]. However, in clinical practice, obstruction by a luminal foreign body is seldom observed and the luminal obstruction seems to be the result rather than the cause of appendicular inflammation [28].

Aravindan demonstrated the histological features of hypersensitivity type I reactions in appendicular specimens of AA and proposed that AA is an allergic reaction [3].

Recent epidemiologic and laboratory studies have shown the presence of an IgE-mediated reaction in AA [4,5,6,7].

Basophils, the least abundant circulating granulocyte [25], are pivotal for the progression and maintenance of allergic inflammation [24]. They have several features that resemble those of MCs, especially since they express receptors that directly bind to IgE and release several mediators, like histamine and Th2 cytokines [24].

Basophils can be recruited into the tissues during inflammatory or immune responses [23].

We found a marginal difference in basophil serum levels within the three study groups, with the highest levels detected in the control group. We hypothesised that, if basophils are mobilised to the appendix, their serum levels will fall as a consequence of this recruitment. Interestingly, this was also observed upon allergen testing in broncho alveolar lavage (BAL). An increase in basophil percentage in BAL negatively correlates with basophil percentage in the blood, thereby indicating the recruitment of basophils from the circulation into the lung [24].

Concerning the serum levels of eosinophils, a marginal significant difference was found, with the highest levels found in the APA group. This may corroborate the hypothesis of an allergic component being involved in AA, since blood eosinophilia is a feature of allergy [29].

As expected, in AGA, high WBCs and neutrophil levels were present. Increased WBC and neutrophil count are the first signs of inflammation in AA. Nevertheless, the sensitivity and diagnostic values vary broadly [30]. Tsuji found the presence of a lymphoplasmocitary infiltrate in the lamina propria of the appendicular wall of AA specimens, suggesting that this presence would be a consequence of stimulation by an endoluminal antigen, which has not yet been clarified [31,32].

MCs have a widespread distribution at strategic locations in nearly all human tissues [14] and act as the key effector cells in the pathogenesis of IgE-mediated allergic diseases [8,33]. The intestine comes into contact with thousands of antigens [33]. The majority are tolerated, but, hypothetically, some can induce a Th2 response. IgE-specific antibody binding to antigens and crosslinking with receptors at the MC surface induces the release of several mediators, including tryptase [8,13,14,15].

Serum tryptase determination is one of the most important tests in the field of allergies [34]. Tryptase is the main protease stored in human MC granules [10]. Although its biological function has not yet been fully clarified, it is well demonstrated that tryptase has a central role in inflammatory reactions and in the immediate allergic reactions initiated by IgE [35]. As tryptase is a specific marker of secretory granules, its presence in various body fluids is a good hallmark of MC activation and degranulation in acute allergic events [35].

Tryptase is found in the nasal secretions of allergic individuals during the pollen season, in urticaria lesions, in intestinal fluid after food challenge, and in blood during anaphylactic events [36].

The concept of the ALF test was inspired by the BAL fluid clinical test. Elevated tryptase activity has been found in BAL fluid related to anaphylaxis and bronchial asthma [10,37].

In our study, a striking difference was found between the control and AA groups, with higher levels being found in AA, whether its presentation was APA or AGA, thus supporting the presence of an allergic component in AA, as we have demonstrated [5,6,7,11].

These differences were present in both ALF and serum. In serum, tryptase levels were 6.5 times higher in APA and 11 times higher in AGA when compared with the values of the control group. For ALF, the differences were larger, with tryptase levels being 109 times higher for APA and 114 times for AGA than the control group, which is compatible with a local allergic reaction.

These results are consistent with a previously proposed hypothesis that allergic responses play a role in the pathologic process of AA [3]. In a previous study, we showed that Th2 cytokines IL-4, IL-5 and IL-9 were elevated in the ALF of patients with AA, which supports the presence of an allergic component in AA [5]. Our group also found an elevation of IgE fixed cells in AA compared with the control group [6] and an elevation of cationic eosinophilic proteins in both the serum and ALF of patients with AA [7]. A correlation was also demonstrated between IL-5 serum and ALF levels and eosinophil infiltration in the appendicular wall [11].

In asthmatic individuals, histamine-induced bronchial smooth muscle contraction is enhanced by tryptase [10]. Tryptase induces degranulation of peripheral blood eosinophils and the release of eosinophil peroxidase and also histamine [4,38]. Tryptase is also chemotactic to MCs [37]. All these events associated with tryptase activity contribute to allergy.

The discriminative power of tryptase serum concentration between the AA and non-AA samples in this study was good, with its AUC being 0.825 [39]. This result, combined with clinical evaluation and other laboratory and imaging diagnostic tests, can be useful for the clinical decision-making process [40].

Conservative treatment of phlegmonous appendicitis is an option [41]. Tryptase levels could be useful for distinguishing phlegmonous from gangrenous appendicitis, but unfortunately the discriminative power of serum tryptase for differential diagnosis between APA and AGA was weak (AUC = 0.559).

Tryptase levels were significantly elevated in patients with peritonitis. However, the diagnosis of AA is clinical and so this elevation is not relevant in the clinical context [42]. In ALF, tryptase levels were also higher in those with peritonitis, but again, for clinical decision making, the significance is residual, as appendectomy has already been performed.

Histamine is the most prominent mediator in allergies [15]. In 1929, Seyle demonstrated that intravenous histamine injection in rats provoked APA without other organs being affected [43].

Muscle contraction is a feature of allergy [44]. Histamine contracts inflamed and normal muscle appendices [45]. Serotonin also induces muscle contraction and mucus production which can obstruct appendicular lumen and is followed by the well-known sequence of events in AA [15,20,45,46]. Local serotonin release exacerbates intraluminal secretion, venous engorgement, vasoconstriction and smooth muscle contraction in the appendix [47].

In our study, histamine was present in all ALF samples, which is not surprising as histamine is present in all tissues [48,49]. We found no difference in histamine levels between patients with AA and those in the control group. However, histamine changes are transitory and may diminish within minutes of release (35, 50); therefore, we speculate that increased histamine from mast cell degranulation may have been unmeasurable in our AA samples due to processing time needed before analysis. A 24 h urinary collection for histamine metabolites may be useful. Such metabolites can be elevated for as long as a day [50].

Concerning serotonin, no difference was found between AA and the control group, neither at local nor at systemic levels.

In this pioneer work, we had to deal with the uncertainty related to the fact that the half-life of serotonin is unknown. We do not know if a delayed time of collection of ALF and serum for analysis of serotonin levels may have affected the residual values that were obtained during the study, as many hours could have passed since the initiation of the AA pathologic process.

A study from another group demonstrated that the serum levels of serotonin decrease over time during the evolution of clinical presentation [51]. In our study, the median times from the onset of clinical symptoms and blood collection were 36 and 47 h, respectively, for APA and AGA; during this gap of time, it is possible that most of the serotonin in serum had already been degraded or eliminated.

Conceptually, histamine and serotonin can participate in AA pathogenesis; however, in the present study histamine and serotonin levels in both ALF and serum were not higher than in the control group, possibly due to the aforementioned reasons.

MCs contribute to various forms of allergy [15] and are involved in IgE-mediated acute allergic cutaneous responses and atopic dermatitis [12]. Their role in asthma has been extensively studied [13]. MCs are also involved in digestive diseases like eosinophilic esophagitis [13] and food allergies [15]. Is AA another allergic disease of the digestive tract?

Strengths of this study: Clear histologic confirmation of AA. To our knowledge, this is the first report of the evaluation of mast cell proteins in ALF and serum in patients with a clinical diagnosis of AA. We evaluated both the local and the systemic inflammatory mast cell response by determining the concentration of mast cell proteins. We characterised ALF components since they reflect the inner composition of pathologic appendices, and for that reason we consider this method a unique, innovative concept for studying local immune–inflammatory reactions in AA.

Weaknesses of this study: The results are provided by a single institution with a relatively small number of patients and no specific forms of tryptase were evaluated. Moreover, mast cell protein determination is usually performed in biological fluids and has never been performed in NaCl 0.9%. Finally, the control group consisted of patients with underlying malignancy and their ages were significantly different from those of the AA population. However, a better control group was not available. Patients operated on overnight were excluded, which may have introduced some bias.

Medical literature is scarce on this topic which warrants further research.

## 4. Materials and Methods

### 4.1. Patients and Study Design

In a previous prospective study, we collected and analysed data on histamine, serotonin and tryptase levels in the ALF and peripheral blood of patients with AA.

All consecutive patients over the age of 18 admitted to the emergency department with a clinical framework-compatible diagnosis of AA and referred for appendectomy between August 2021 and April 2022 were eligible for this study.

Patients under 18-year-old were excluded because these patients were admitted to the paediatrics department. Pregnant women were also excluded.

Patients who underwent appendectomy between 9 PM and 8 AM were also excluded because during this period there was no possibility to process the samples at the laboratory for MC protein determination.

Diagnoses of AA were confirmed by histology.

Patients who underwent appendectomy for right iliac fossa pain with no histological criteria of appendicitis were to be included in the control group; however, a mid-term evaluation showed that all patients undergoing appendectomy had histological criteria for appendicitis, which in practice made the existence of a control group unfeasible. Incidental appendectomy is a rarely performed procedure; therefore, the time taken to select a control group would have not been feasible. A right colectomy is a common procedure for colonic cancer and thus the control group was extended to patients with colonic neoplasia who underwent right colectomy [52]. Appendectomy was performed on colectomy specimens and ALF collection was carried out. A control group of patients with no histology criteria for appendicitis was enrolled [52].

### 4.2. Pathologic Analysis

After appendectomy, all specimens were fixed in 10% formalin. Three cross sections were taken from each specimen representing the base, the middle and the tip of the appendix.

They were routinely processed, paraffin embedded and stained with haematoxylin and eosin (H&E) [53]. All the appendicular specimens were evaluated by one of the authors (CH), a dedicated gastrointestinal pathologist who was purposefully blinded to the clinical picture and initial histologic report.

The most important criterion for AA is the infiltration of polymorphonuclear neutrophils in the *muscularis propria* [54,55,56].

Acute phlegmonous or suppurative appendicitis (APA) was defined as neutrophilic infiltration in the *muscularis propria* and acute gangrenous appendicitis (AGA) was defined as necrosis of the wall of the appendix with a background of transmural inflammation [28].

Neutrophil presence in the mucosa was considered a normal variant with no clinical significance [56]. The specimens were classified as normal when no neutrophil infiltration was shown in the *muscularis propria* [28,57].

### 4.3. Laboratory Procedure

#### Hemogram

All the patients included in the study had blood samples collected on admission to the emergency department and measured by an automated DxH 900 haematology analyser (Beckman Coulter, Inc., Pasadena, CA, USA) using the Coulter principle and standards. Concerning white blood cell (WBC) population count for differential testing, VCSn technology was used.

The results were automatically expressed in mm^3^ in peripheral blood (Sysmex, XS-800i) (Beckman Coulter, Inc., Pasadena, CA, USA). The count was made in triplicate to ensure the accuracy and reproducibility of the results.

The reference interval for WBC count was 4.0–11.0 × 10^9^/L, that for neutrophils was 1.90–8.00 × 10^9^/L (40–70%), that for lymphocyte was 0.9–5.20 × 10^9^/L (19–48%), that for basophils was 0.00–0.20 × 10^9^/L (0.0–1.5%), that for eosinophils was 0.00–0.80 × 10^9^/L (0.0–7.0%) and that for monocytes was 0.16–1.00 × 10^9^/L (3.4–9.0%).

### 4.4. Appendicular Lavage Fluid

After appendectomy, a gauge was inserted in the caecal side of the lumen of the appendix and 3 mL of saline 0.9% was administered and collected, and re-administered and re-collected 3 additional times.

Previous training was carried out for appendicular lavage so that the process was standardized and data collection uniformised. A tutorial was presented to all the surgeons that performed an appendectomy. The appendicular fluid samples were collected with Sarstedt Monovette tubes and centrifuged. A total of 1 mL of the supernatant was extracted and stored at −20 °C [5].

### 4.5. Mast Cell Protein Determination

Blood samples for mast cell protein determination were collected immediately before the induction of anaesthesia. Plasma was obtained from EDTA-anticoagulated blood, kept at 2–8 °C and centrifugated within one hour after phlebotomy.

ALF was also kept at 2–8 °C and centrifuged at 3000 rpm as quickly as possible after collection. Only the supernatant as described above was kept for analysis.

Serum, plasma and ALF were aliquoted, stored at –70 °C and thawed once just before assaying for mediator levels.

MC activation releases a wide variety of pro-inflammatory mediators, such as histamine, tryptase and serotonin [8,13,14,15]. These pro-inflammatory mediators are associated with a Th2 reaction and were obtained from the Joaquim Chaves Lab. For the determination of histamine and serotonin levels, plasma and ALF were assayed using a commercial ELISA ki, according to the manufacturer’s instructions (DRG Instruments GmbH, Marburg, Germany). Serum tryptase levels were measured using the ImmunoCAP System (Thermo Fisher Scientific Inc., Waltham, MN, USA). ImmunoCAP Tryptase was used to measure total tryptase levels, including all forms of α-tryptase and ß-tryptase [34].

The reference values for histamine [58,59,60] were 0.2–1.0 ng/mL, those for serotonin [61,62,63] were 30–200 ng/mL and those for tryptase [34,64,65] were <11.4 ng/mL. No previous reference values for ALF concentrations were available.

### 4.6. Other Definitions

The presence of gangrenous alterations with a transmural defect at the appendicular wall or a loose fecalith in the abdominal cavity defined appendicular perforation [66,67].

On clinical examination, peritonitis is identified as rigidity, rebound tenderness or guarding at the abdominal palpation [42]. Peritonitis was defined as exudative inflammation in the peritoneum by the operating surgeon and qualified as localized peritonitis when one or two quadrants were involved or as generalised peritonitis when more than two quadrants were involved [68].

Complicated appendicitis was defined when any of the following criteria were present: perforation, gangrenous appendicitis (histological criteria), peritonitis, appendicular plastron or pelvic or intra-abdominal abscess [69,70].

Uncomplicated appendicitis was defined as an inflamed appendix in the absence of gangrene, perforation or abscess around the appendix [69].

### 4.7. Ethics

This study was approved by the Ethics Committee of the Hospital Garcia de Orta (Centro Garcia de Orta, reference number 82/2021; date of approval: 23 June 2021). The study did not interfere with any clinical decisions or pose any risk for patients. The information obtained may have an impact in the future management of patients’ treatment, particularly in considering other therapeutic alternatives.

As described above, an interim analysis showed no patients could be recruited for the control group; thus, an addendum was requested and approved (addendum 82/2021; date of approval: 11 November 2021) to include the control group patients referred for right colectomy.

Written informed consent was obtained from all the patients after having been exhaustively informed about the objective of the study. All data were pseudo-anonymised and the results were presented in such a way that made it impossible to identify single patients.

### 4.8. Statistical Analysis

Data were analysed using R statistical software V4.2.2 for Windows.

The results were considered significant at the 5% significant level. To test the normality of the data, the Shapiro–Wilk test was used. For sample characterisation, frequency analysis (*n*, %) was used for qualitative data; mean ± standard deviation or median—percentile 25% to percentile 75% (Q1–Q3) were used for quantitative data, according to the normality of the data.

To compare clinical presentations between appendicular histology (from two study groups, APA and AGA) and to compare peripheral blood mast cell granule proteins and ALF mast cell granule proteins according to clinic presentation (perforation, peritonitis), the Mann–Whitney test was used since the normality assumption had not been verified.

To compare age, BMI, hemogram results, mast cell granule protein blood concentrations and mast cell granule protein ALF concentrations between the appendicular histology categories (three study groups: control, APA and AGA), the one-way ANOVA (if normality was verified) or the Kruskal–Wallis test (if normality of the data was not verified) were used. When statistically significant differences were detected using the one-way ANOVA or Kruskal–Wallis test, Tukey–HSD or Kruskal–Wallis multiple comparison tests were used, respectively.

To compare the time between initiation of clinical symptoms and appendectomy, tryptase, histamine and serotonin concentrations, perforation, peritonitis and complicated appendicitis, the *t* test (if normality was verified) or Mann–Whitney test (if normality was not verified) were used.

To study the relationship between two qualitative variables, the Chi-squared test or the Chi-squared test by Monte Carlo simulation were used, depending on whether the applicability assumptions of the Chi-squared test had been verified.

To study the relationship between two quantitative variables, the Spearman correlation coefficient was used since the normality assumption had not been verified.

ROC analysis was performed to evaluate tryptase serum levels for the diagnosis of acute appendicitis and to distinguish APA from AGA and the diagnosis of peritonitis and perforation. The AUC varied between 0 and 1. The discriminant power was classified: 0.5—no discriminative; [0.5, 0.7]—weak; [0.7, 0.8]—acceptable; [0.8, 0.9]—good and ≥0.9—excellent [39].

### 4.9. Other Data

Personal information including previous similar episodes, surgical details, open or laparoscopic appendectomy, complications, hospital stay and other histologic data were evaluated. Individuals were also asked about any symptoms pertaining to allergic disorders. Ultrasonography was performed in most of the patients for diagnosis and, in some of them, CT abdomino-pelvic scans were performed. These details are relevant has they provided more information on the study groups.

## 5. Conclusions

Tryptase is associated with a hypersensitivity type I reaction. In the present study, tryptase levels were markedly elevated in the serum and appendicular lavage fluid of patients with acute appendicitis, consistent with mast cell degranulation

A key step for triggering allergic inflammation in the skin, lung and intestine is mast cell degranulation. It seems that the same happens in the appendix. The data from this study support our previous hypothesis that allergic inflammation may also be an aetiologic factor in AA. These findings may have potential clinical relevance in AA treatment and need further investigation.

## Figures and Tables

**Figure 1 ijms-25-01645-f001:**
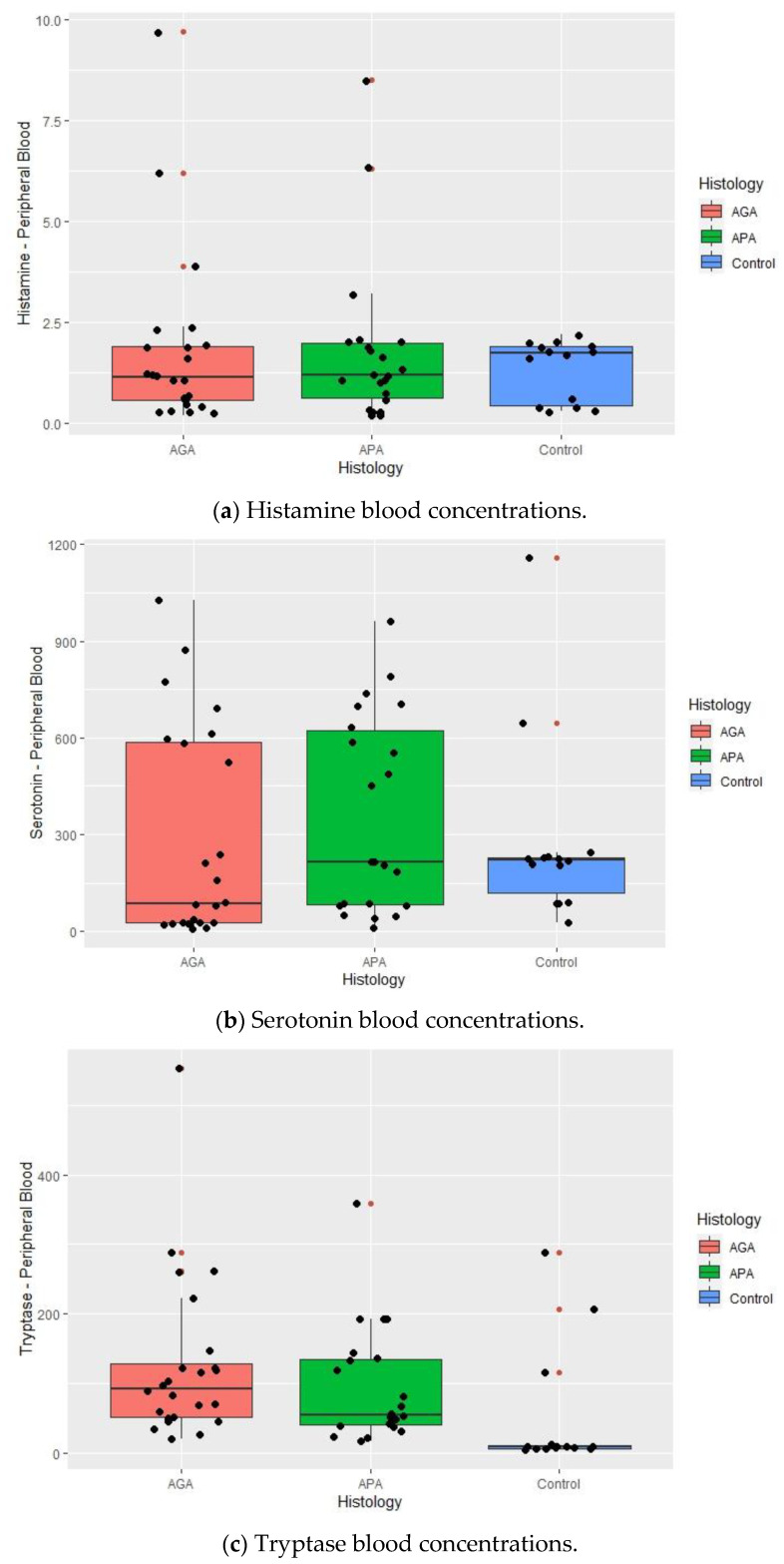
Boxplots summarizing histamine (**a**), serotonin (**b**) and tryptase (**c**) blood concentrations in the normal appendix (control), acute phlegmonous (APA) and gangrenous appendicitis (AGA) groups. Median values and interquartile ranges are denoted by horizontal bars and boxes. Outliers are represented by • and observed values by •.

**Figure 2 ijms-25-01645-f002:**
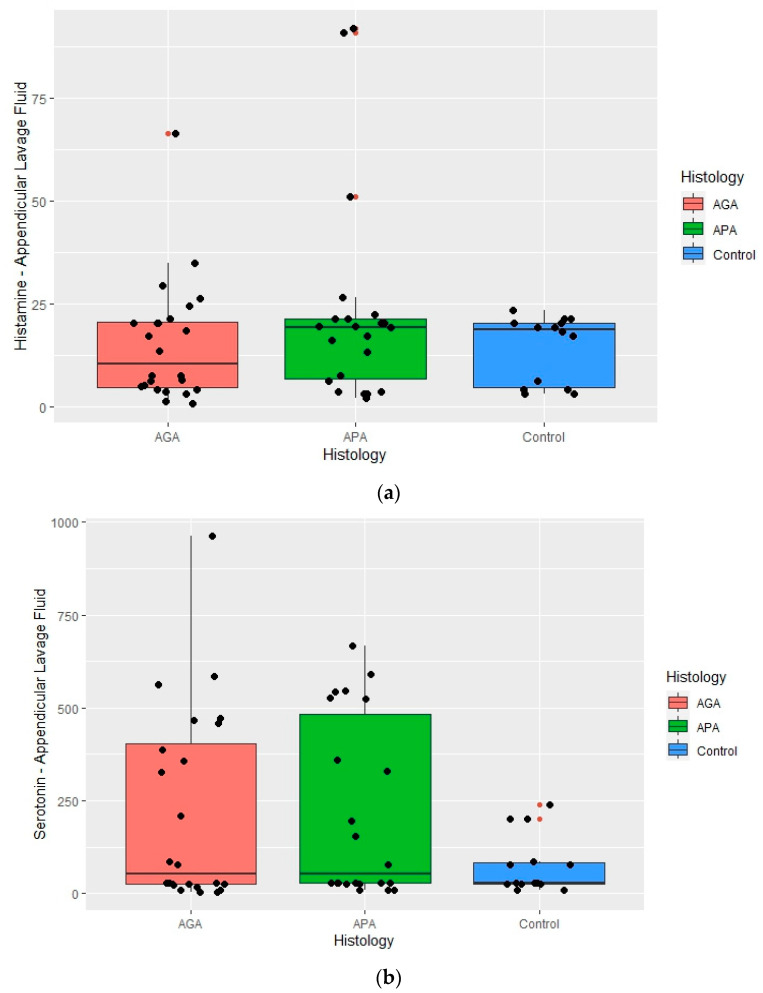
Boxplots summarizing histamine, serotonin and tryptase appendicular lavage fluid concentrations in the normal appendix (control), acute phlegmonous (APA) and gangrenous appendicitis (AGA) groups. Median values and interquartile ranges are denoted by horizontal bars and boxes. Outliers are represented by • and observed values by •. (**a**) Histamine appendicular fluid lavage concentrations. (**b**) Serotonin appendicular fluid lavage concentrations. (**c**) Tryptase appendicular fluid lavage concentrations.

**Figure 3 ijms-25-01645-f003:**
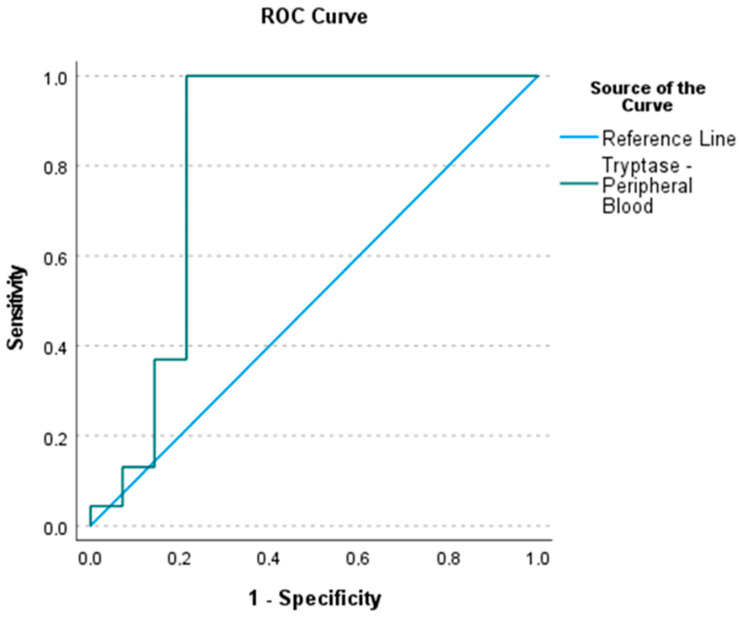
Receiver operating characteristic (ROC) curves of tryptase serum levels for the diagnosis of acute appendicitis. The area under the curve (AUC) for tryptase serum levels, being the optimal sum of sensitivity (1.00) and specificity (0.214) at a cut-off level of values greater than 13.95 ng/mL, is 0.825 (*p* < 0.001).

**Table 1 ijms-25-01645-t001:** Study groups and patient demographics.

	Control	APA	AGA	
*n* (%)	14 (23.3)	22 (36.7)	24 (40)	*p* Value
Age (y)	67.5 (30–78)	48 (36–66)	31 (23–55)	**0.044 ***
Sex M/F	9/5 (64.3/35.7)	11/11 (50/50)	12/12 (50/50)	0.644 **
Allergy N/Y	12/1 (92.3/7.7)	18/2 (90/10)	20/1 (95.2/4.8)	0.834 ***
BMI	28.04 ± 4.76	24.57 ± 4.07	26.44 ± 4.540	0.298 ****
Clinics	-	36.02 (29–41)	47 (37–70)	0.130 *****

APA—acute phlegmonous appendicitis; AGA—acute gangrenous appendicitis. M—male; F—female. N—no; Y—yes; BMI—body mass index, kg/m^2^. Clinics: time elapsed between the onset first symptoms and appendectomy. Results are presented as mean ± SD and median (Q1–Q3). Sex is presented as a number and %. * Kruskal–Wallis test. ** Chi-squared test. *** Chi-squared test by Monte Carlo Simulation. **** One-way ANOVA. ***** Mann–Whitney test. Statistically significant differences were set at a 5% significance level (**bold**).

**Table 2 ijms-25-01645-t002:** Hemogram and appendicular histology.

	Control	APA	AGA	*p* Value
WBC	10.10 ± 3.43	12.08 ± 2.93	15.97 ± 5.05	**0.000 ****
Neutrophils	6.73 (5.69–9.28)	8.21 (7.20–12.46)	11.50 (9.50–14.70)	**0.005 ***
Lymphocytes	1.29 (1.05–1.86)	1.83 (1.63–2.42)	1.37 (0.83–2.10)	0.253 *
Basophils	0.06 ± 0.02	0.03 ± 0.02	0.04 ± 0.02	0.051 **
Eosinophils	0.04 (0.01–0.21)	0.14 (0.04–0.26)	0.02 (0.00–0.07)	0.052 *
Monocytes	0.06 (0.04–0.07)	0.03 (0.02–0.05)	0.03 (0.02–0.06)	**0.034 ***

APA—acute phlegmonous appendicitis; AGA—acute gangrenous appendicitis; WBC—white blood count. WBC and basophils are expressed in absolute numbers × 10^9^/L, with mean ± SD; neutrophils, lymphocytes, eosinophils and monocytes are expressed in absolute numbers × 10^9^/L, with median (Q1–Q3); * Kruskal–Wallis test. ** One-way ANOVA. Statistically significant differences were set at a 5% significance level (**bold**).

**Table 3 ijms-25-01645-t003:** Mast cell blood granule protein levels and appendicular histology.

	Control	APA	AGA	*p* Value
Histamine	1.8 (0.4–1.9)	1.2 (0.6–2.0)	1.2 (0.6–1.9)	0.948
Serotonin	221.25 (89.5–229.5)	214.25 (80.5–633.12)	87.0 (22.25–589.95)	0.290
Tryptase	8.25 (5.6–11.5)	53.95 (38.77–135.18)	92.58 (50.16–134.26)	**0.001**

APA—acute phlegmonous appendicitis; AGA—acute gangrenous appendicitis. Measurements are presented in pg/mL. Results are presented as medians (Q1–Q3). Kruskal–Wallis test. Statistically significant differences were set at a 5% significance level (**bold**).

**Table 4 ijms-25-01645-t004:** Mast cell ALF granule protein levels and appendicular histology.

	Control	APA	AGA	*p* Value
Histamine	18.73 (4.28–20.33)	19.38 (6.42–21.40)	10.50 (4.60–20.92)	0.756
Serotonin	28.00 (24.5–84.00)	53.43 (26.5–523.95)	52.25 (24.25–422.05)	0.474
Tryptase	7.3 (4.5–10.3)	796.8 (194.1–980.5)	837.4 (272.6–1075.1)	**0.001**

ALF—appendicular lavage fluid. APA—acute phlegmonous appendicitis; AGA—acute gangrenous appendicitis. Measurements are presented in pg/mL. Results are presented as medians (Q1–Q3). Kruskal–Wallis test. Statistically significant differences were set at a 5% significance level (**bold**).

**Table 5 ijms-25-01645-t005:** Mastocyte granule proteins: blood and ALF correlations.

	Hist (ALF)	Ser (PB)	Ser (ALF)	Try (PB)	Try (ALF)
**ρ (*p* Value)**					
Hist (PB)	0.612 * (**0.000**)	−0.047 (0.715)	0.068 (0.602)	0.425 * (**0.001**)	0.078 (0.548)
Hist (ALF)		0.068 (0.602)	0.073 (0.574)	0.177 (0.169)	0.428 * (**0.001**)
Ser (PB)			0.514 * (**0.000**)	−0.177 (0.168)	−0.140 (0.277)
Ser (ALF)				0.034 (0.795)	0.029 (0.826)
Try (PB)					0.684 * (**0.000**)

PB—peripheral blood; ALF- appendicular lavage fluid. Hist—histamine; Ser—serotonin; Try—tryptase. ρ: Spearman correlation coefficient. (*p* value). * Correlation is significant at the 0.01 level (two-tailed). Statistically significant differences were set at a 5% significance level (**bold**).

**Table 6 ijms-25-01645-t006:** Correlation between blood basophils and PB and ALF histamine, serotonin and tryptase.

		Histamine	Serotonin	Tryptase
Basophils	PB	0.461 ** (**0.002**)	−0.275 (0.070)	0.167 (0.280)
Basophils	ALF	0.353 *(**0.019**)	−0.298 * (**0.049**)	−0.28 (0.856)
		ρ (*p* value)		

PB—peripheral blood, ALF—appendicular lavage fluid. ρ—Spearman correlation coefficient. ** Correlation is significant at the 0.01 level (two-tailed) (**bold**). * Correlation is significant at the 0.05 level (two-tailed) (**bold**) (*p* value).

**Table 7 ijms-25-01645-t007:** Peripheral blood mast cell granule proteins and clinical presentation.

	Histamine		Serotonin		Tryptase	
		*p* Value *		*p* Value		*p* Value
Perforation						
No	1.3 (0.6–2.0)	0.371	214.2 (83.5–584)	0.143	156.9 (20.8–135.1)	0.712
Yes	1 (0.4–1.2)		26.5 (23.76–612)		58.91 (45.1–70.2)	
Peritonitis						
No	1.3 (0.6–2)	0.906	214.2 (86.5–539.2)	0.495	45.9 (9.25–119.64)	**0.020**
Yes	4.6 (3.05–5.14)		51.30 (22.2–64.3)		241.6 (168–278.35)	

Results are presented as medians (Q1–Q3). Measurements are presented in ng/mL. * The Mann–Whitney test was used to analyse differences between groups. Statistically significant differences were set at a 5% significance level (**bold**).

**Table 8 ijms-25-01645-t008:** ALF mast cell granule proteins and clinical presentation.

	Histamine		Serotonin		Tryptase	
*n* (%)		*p* Value *		*p* Value		*p* Value
Perforation						
No	17.6 (5.5–21.4)	0.271	52.5 (26–329.7)	0.767	635.6 (120.-956.7)	0.693
Yes	5.6 (1.3–20.33)		117.7 (15.5–457.9)		243.2 (95.1–885.3)	
Peritonitis						
No	17.1 (4.6–20.8)	0.962	28.5 (25–237.7)	0.233	293.1 (8.9–929.3)	**0.033**
Yes	17.8 (5.0–20.9)		140 (26.2–462.25)		87.8 (373.3–992.0)	

ALF—appendicular lavage fluid. Results are presented as medians (Q1–Q3). Measurements are presented in ng/mL. * The Mann–Whitney was used to analyse differences between groups. Statistically significant differences were set at a 5% significance level (**bold**).

## Data Availability

The data presented in this study are available on request from the corresponding author. The data are not publicly available to protect patients’ privacy.

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
