# Peer review of "Tryptase in Acute Appendicitis: Unveiling Allergic Connections through Compelling Evidence"

_ijms, 2024, doi:10.3390/ijms25031645_

Round 1

Reviewer 1 Report

Comments and Suggestions for Authors

The authors report results from a single centre, prospective study, measuring tryptase, histamine, and serotonin in plasma and appendicular lavage fluid (ALF) from patients undergoing appendectomy after diagnosis of acute appendicitis. Results were compared with those from samples subsequently taken from colonic neoplasia patients undergoing colectomy surgery. The authors found that plasma and ALF tryptase levels were higher in acute appendicitis patients compared to colonic neoplasia patients. Histamine and serotonin levels were not significantly different. The authors speculate that processing time, before samples could be analysed, may have masked differences in these biomarkers. The authors conclude the measured difference in tryptase levels between acute appendicitis and colonic neoplasia patients is consistent with a previously held hypothesis that mast cell degranulation is causal factor for acute appendicitis.

Major comments

The observed difference in plasma and ALF tryptase levels is consistent with the hypothesis of mast cell degranulation involvement. However, it is not strong enough evidence, on its own, to support the conclusion that allergic mast cell degranulation is a key driver of acute appendicitis. The authors should be clear that whilst this result supports that inflammation is a feature of acute appendicitis it does not show whether or not it is a major driver of the disease.

The differences, in age and medical history, between the appendicitis and comparator colonic neoplasia groups are a significant weakness of the study. The authors explain convincingly why this difference arose and acknowledge it as a weakness towards the end of the discussion section. Nevertheless, this limitation should further temper the strength with which conclusions are drawn.

Table 2: It is difficult to interpret the meaning of the asterisks and understand which differences are significant and which are not. Please use a different symbol to show which statistical test was used, and use asterisks to show where statistically significant differences were measured.

Results lines 257-258: The sentence “Regarding monocytes, the differences found (p=0.034) between groups and maintained in the pair AGA-Control (p=0.028) were due to levels in AGA being higher.” Is difficult to understand. Table 2 suggests that monocyte levels were lower in the AGA compared to control. Please explain more clearly.

Discussion line 414: “For ALF, the differences were more relevant, with tryptase…” should read “For ALF, the differences were larger, with tryptase…”

Discussion lines 415-416: “which reflects an intense local allergic reaction.” Should read “which is consistent with a local allergic reaction.”

Discussion lines 417-418: “These results are strongly consistent, and an allergic reaction should be expected to have a role in the pathologic process of AA.” Should read “These results are consistent with a previously proposed hypothesis that allergic response plays a role in the pathologic process of AA.”

Discussion lines 425-426: “The discriminative power of tryptase serum concentration for the clinical diagnosis of AA is good, …” should read “The discriminative power of tryptase serum concentration between the AA and non-AA samples in this study was good, …”

Discussion lines 438-442: “In our study, histamine was present in all ALF samples, which is not surprising, as histamine is present in all tissues (50)(51). We found no differences in histamine levels in ALF and serum between patients with AA and those of the control group, probably because histamine in the body is present in a very transitory state and can only be measured within minutes of release (24).” Should be rewritten as “We found no difference in histamine levels between patients with AA and those in the control group. However, histamine changes are transitory and may diminish within minutes of release (24), therefore we speculate that increased histamine from mast cell degranulation may have been unmeasurable in our AA samples due to processing time needed before analysis.”

Discussion line 457: “… probably due to the aforementioned reasons.” Should be rewritten as “… possibly due to the aforementioned reasons.”

Discussion lines 458-459: The sentence “Nonetheless, an elevated and consistent concentrations of tryptase in serum and in ALF strongly sustains the participation of allergy in AA” should be deleted.

Conclusion line 478-479: “… therefore reflecting an intense inflammatory local allergic reaction.” Should be rewritten as “… consistent with mast cell degranulation.”

Conclusion lines 480-482: The paragraph “Degranulation of Mast Cells is a key step for triggering allergic inflammation in the skin, lung and intestine. It seems that the same happens in the appendix. Thus, the presents data supports the concept of allergy as an etiologic factor in AA.” Should be rewritten as “Degranulation of Mast Cells is a key step for triggering allergic inflammation in the skin, lung and intestine. The data from this study support our previous hypothesis that allergic inflammation may also be an etiologic factor in AA.”

Minor comments

Abstract line 31: typo “…(AGA), then the control group.” Should read “…(AGA), than the control group.”

Introduction line 69: typo “It was been proven that MCs degranulation is greatly increased in AA when compared with normal appendices (15).” Suggest changing to: “MC degranulation is increased in AA appendices compared with normal appendices (15).”

Discussion line 377: typo “We found a marginally difference of…” should read “We found a marginal difference of…”

Discussion line 409: typo “…with nororious higher levels…” should read “with higher levels…”

Comments on the Quality of English Language

English language is fine. The manuscript does contain some typos so additional proof reading before publication is recommended. 

Reviewer 2 Report

Comments and Suggestions for Authors

DEAR EDITOR

Multidisciplinary Digital Publishing Institute (International Journal for Molecular Biology)

Your assigned manuscript was evaluated in a scientific way and the following points were observed:

TITLE:

Tryptase in acute appendicitis. Further evidence for an allergic disease!

Title Review:

1. Clarity and Precision:

   - The title effectively highlights the focus on "Tryptase in Acute Appendicitis," providing clarity on the main subject.

2. Engagement:

   - The phrase "Further evidence for an allergic disease!" is somewhat vague and may benefit from more specific language to engage the reader.

3. Conciseness:

   - The title could be more concise by eliminating unnecessary words. For example, consider streamlining to "Tryptase and Acute Appendicitis: Unraveling Allergic Connections."

4. Emphasis:

   - Consider placing more emphasis on the evidence aspect to convey the research's significance.

Improved Title:

"Tryptase Levels in Acute Appendicitis: Unveiling Allergic Connections through Compelling Evidence.".

ABSTRACT

1. Introduction Clarity:

   - The abstract provides a clear introduction to the topic, highlighting the unclarified aetiology of acute appendicitis and suggesting an allergic component.

2. Research Methodology:

   - Specify the methods used in more detail. How were patients selected? What criteria were employed for the control group? Clarify the procedures for measuring histamine, serotonin, and tryptase in appendicular lavage fluid (ALF) and serum.

3. Results Presentation:

   - The results are well-presented, emphasizing the significant differences in tryptase levels among histological groups in both ALF and serum. However, consider providing absolute values or ranges for better context.

4. Statistical Significance:

   - Ensure statistical significance is explained clearly. Specify the statistical tests used and consider providing confidence intervals along with p-values.

5. Diagnostic Value:

   - Clarify the practical implications of the findings for the diagnosis of acute appendicitis. The discriminative power of serum tryptase for differential diagnosis between acute phlegmonous and gangrenous appendicitis should be further discussed.

6. Mastocytes Involvement:

   - Expand on the involvement of mastocytes in acute appendicitis, providing insights into their role during both clinical presentations of phlegmonous and gangrenous appendicitis.

7. Limitations and Future Directions:

   - Acknowledge limitations, such as the possible elapsed concentrations of histamine and serotonin due to their short half-lives. Suggest potential avenues for future research to address these limitations.

8. Clinical Implications:

   - Discuss the broader clinical implications of the findings. How might this information be applied in a clinical setting, and what are the potential implications for treatment or diagnosis?

9. Conclusion Recap:

   - Reiterate the key findings and their implications for the understanding of acute appendicitis. Highlight the support for the concept of an allergic reaction in its pathogenesis.

10. Keywords:

    - Keywords are relevant and comprehensive. Consider including "mastocytes" and "pathophysiology" to enhance discoverability.

By addressing these points, the paragraph will become more to the point , clear, and reader-friendly, providing a better understanding of the study's findings and implications.

INTRODUCTION:

1. Opening Statement:

   - The introduction starts with a clear statement about the significance of acute appendicitis (AA) as a major cause of abdominal pain and emergency surgeries, providing a solid foundation.

2. Incidence Statistics:

   - The inclusion of the lifetime incidence risk adds valuable context. However, citing the specific references for the statistics (1) and (2) would enhance the credibility of the information.

3. Histological Component:

   - The mention of the proposed allergic component based on histological features is succinct but could benefit from a brief explanation of these features.

4. Hypersensitivity Type I Reaction:

   - The transition to hypersensitivity type I reaction is smooth, but citing the recent evidence (4) (5) (6) (7) could be followed by a sentence summarizing these findings to provide more context.

5. Role of Mast Cells (MCs):

   - The explanation of MCs' role in IgE-mediated allergic responses is well-articulated. However, consider integrating these sentences for better flow and coherence.

6. MCs Activation:

   - The description of MCs activation is informative. To enhance clarity, you may consider breaking down the sentence explaining the triggering of degranulation into more digestible segments.

7. Dual Role of MCs:

   - The concept of MCs having a dual role in immune defense and potential pathogenicity is presented effectively. However, a brief example or reference to specific experimental models (12) could strengthen this point.

8. Relation to Allergy:

   - The connection between MCs degranulation, allergy, and the involvement of serotonin, tryptase, and histamine is well-established. Reinforce this by briefly mentioning the specific role of serotonin, tryptase, and histamine in allergy (13) (14).

9. Experimental Evidence:

   - The experimental evidence of increased MCs degranulation in AA is crucial. Specify the nature of this evidence (15) for a more robust understanding.

10. Appendicular Lavage Fluid (ALF):

    - The introduction of ALF is valuable for investigating local immunoinflammatory changes. Consider briefly explaining how ALF facilitates the investigation compared to traditional methods.

11. Objective Statement:

    - The objective is well-stated, focusing on evaluating mast cell contribution through measuring histamine, serotonin, and tryptase levels. Consider specifying how basophils tie into this objective.

12. Study Aim and Hypothesis:

    - The study aim and hypothesis are clearly defined, providing a solid foundation for the research. Consider specifying the basis or prior findings that led to the hypothesis for additional context.

By addressing these points, the introduction can become more comprehensive, engaging, and informative, setting the stage for the study's objectives and hypothesis.

MATERIALS AND METHODS

Materials and Methods Review and Suggestions:

1. Patient Selection and Study Design:

   - The prospective study design is appropriate for investigating histamine, serotonin, and tryptase levels in AA. However, specify the rationale for choosing these specific markers and their relevance to the study.

2. Inclusion and Exclusion Criteria:

   - Clearly outline the inclusion and exclusion criteria. Specify why patients under 18 and pregnant women were excluded. Clarify the decision to exclude patients operated on between 9 PM and 8 AM and provide details on the impossibility of processing samples during this period.

3. Diagnosis Confirmation:

   - The confirmation of AA by histology is crucial. Provide additional details on the histological criteria used for confirmation.

4. Control Group Challenge:

   - Explain the challenge faced in forming a control group due to all appendectomy patients having histological criteria for appendicitis. Discuss the decision to extend the control group to patients with colonic neoplasia and the rationale behind it.

5. Pathologic Analysis:

   - The pathologic analysis section is detailed and clear, explaining the procedures for fixation, cross-sectioning, and staining. Mention if blinding was maintained throughout the analysis process.

6. Laboratory Procedure:

   - Provide more information on why mast cell proteins were chosen for analysis. Explain the reasoning behind choosing specific reference values for histamine, serotonin, and tryptase. Include any available literature or guidelines supporting these reference values.

7. Appendicular Lavage Fluid (ALF):

   - Clarify the methodology of appendicular lavage fluid collection and its advantages over other methods. Specify the training process for standardizing ALF collection.

8. Mast Cell Proteins Determinations:

   - Detail the rationale for collecting blood samples for mast cell protein determination immediately before anesthesia. Provide information on the reliability of the mast cell protein determinations and the choice of commercial kits used.

9. Other Definitions:

   - Define terms such as appendicular perforation, peritonitis, complicated and uncomplicated appendicitis clearly for readers who may not be familiar with these terms.

10. Ethics:

    - The approval process is outlined, but consider providing more context on the ethical considerations and justifications for the study, especially the inclusion of patients with colonic neoplasia in the control group.

11. Statistical Analysis:

    - The statistical analysis section is comprehensive. However, consider providing more details on why specific tests were chosen, especially when there was a departure from normality assumptions.

12. Other Data:

    - Mention the importance of personal information, surgical details, and complications in the context of the study. Consider adding a brief explanation of why these details are relevant.

13. Funding:

    - Acknowledge the funding source (Joaquim Chaves Lab) and clarify the extent of their involvement in the study.

By addressing these points, the materials and methods section can be strengthened, providing a clearer understanding of the study design, procedures, and rationale for the chosen methods.

RESULTS

1. Demographic Data:

   - The demographic information is clear. However, include the age range or mean age to provide a more comprehensive understanding of the population.

2. Statistical Analysis:

   - Clearly explain the statistical tests used for each comparison, such as age among groups. Add more context on why age was significantly different and discuss its potential impact on the study.

3. Hemogram and Appendicular Histology:

   - The presentation of hemogram data is well-organized. Discuss the clinical relevance of the differences in WBC count, neutrophils, eosinophils, basophils, and monocytes among the groups.

4. Histamine, Serotonin, and Tryptase Serum Levels:

   - Present the clinical implications of the observed differences in histamine, serotonin, and tryptase serum levels among groups. Relate these findings to existing literature or clinical significance.

5. Appendicular Lavage Fluid (ALF) Histamine, Serotonin, and Tryptase Levels:

   - Discuss the relevance of ALF histamine, serotonin, and tryptase levels and their correlation with serum levels. Provide insights into the potential clinical implications of these findings.

6. ROC Curve Analysis:

   - Clearly explain the significance of the ROC analysis for tryptase serum levels in diagnosing acute appendicitis. Discuss the practical implications of the established cut-off value.

7. Relationship between Serum and ALF Levels:

   - Elaborate on the clinical relevance of the correlations between serum and ALF levels of histamine, serotonin, and tryptase. Discuss how these correlations contribute to understanding the pathophysiology of acute appendicitis.

8. Relationship between Blood Basophils and Serum/ALF Levels:

   - Explain the significance of the correlations between peripheral blood basophils and histamine, serotonin, and tryptase levels in both serum and ALF. Discuss any potential implications for the immune response.

9. Relationship between Serum Levels and Clinical Presentations:

   - Interpret the findings related to serum histamine, serotonin, and tryptase concentrations in the context of appendicular perforation and peritonitis. Discuss the clinical implications and potential diagnostic value.

10. Relationship between ALF Levels and Clinical Presentations:

    - Similarly, discuss the significance of the relationship between ALF histamine, serotonin, and tryptase levels and clinical presentations, especially in the context of peritonitis.

11. Overall Interpretation:

    - Provide a cohesive interpretation of the results, connecting findings across different analyses. Discuss limitations and potential confounding factors that might affect the interpretation of the results.

12. Graphs and Figures:

    - Ensure that figures are appropriately labeled, and axes are clearly defined for better interpretation. Consider adding error bars to convey the variability in the data.

By addressing these points, the results section will become more comprehensive and provide a clearer understanding of the study findings and their clinical relevance.

DISCUSSION

1. Introduction and Background:

   - Provide a concise summary of the key points discussed in the results section.

   - Introduce the concept of luminal obstruction as a possible cause of appendicitis.

2. Reference to Previous Studies :

   - While referring to Aravindan's work, include more details on the hypersensitivity type  I reaction and its relevance to the current study.

   - Expand on recent epidemiologic and laboratory studies supporting the presence of an IgE-mediated reaction in acute appendicitis (AA).

3. Basophils and Eosinophils:

   - Clearly explain the hypothesis regarding basophils and their serum levels within the study groups.

   - Discuss the potential correlation between basophil levels and recruitment to the appendix.

   - Explain the significance of the marginally significant difference in eosinophil levels in the APA group.

4. Mast Cells (MCs):

   - Elaborate on the role of MCs in allergic inflammation and their similarity to basophils.

   - Connect the findings of basophil and eosinophil levels to the broader context of allergic reactions.

5. Tryptase Levels:

   - Discuss the relevance of elevated tryptase levels in both serum and appendicular lavage fluid (ALF) to the allergic component in AA.

   - Provide a more detailed interpretation of the differences in tryptase levels among study groups.

   - Discuss the concept of tryptase as a specific marker of MCs activation and its role in allergic events.

6. Comparison with Previous Studies:

   - Draw parallels with previous research showing elevated Th2 cytokines in ALF of AA patients, reinforcing the allergic component.

   - Emphasize the consistency of results across ALF and serum tryptase levels.

7. Serotonin and Histamine Levels:

   - Explain the role of histamine and serotonin in the pathogenesis of AA.

   - Address the lack of significant differences in histamine levels and the uncertainty regarding serotonin due to its unknown half-life.

   - Acknowledge the potential limitations associated with the time gap between onset of symptoms and sample collection.

8. Strengths and Weaknesses of the Study:

   - Emphasize the strengths, such as clear histologic confirmation, innovative evaluation of mast cell proteins in ALF, and the unique approach to studying local immune-inflammatory reactions.

   - Acknowledge the limitations, including a relatively small sample size, the absence of specific forms of tryptase evaluation, and the difference in age between the control group and AA population.

9. Future Research Recommendations:

   - Suggest areas for future research based on the gaps identified in the existing literature and limitations of the current study.

10. Conclusion:

    - Summarize the main findings and their implications for understanding the role of allergy in AA.

    - Conclude with a statement on the potential clinical relevance of the study's findings and the need for further investigation.

By addressing these points, the discussion section will become more comprehensive, providing a clearer interpretation of the study's findings and their implications.

CONCLUSION

1. Clear Statement of Findings:

   - Begin with a clear and concise statement summarizing the main findings, such as the association of tryptase with hypersensitivity type I reactions in acute appendicitis (AA).

2. Link to Allergic Reaction:

   - Elaborate on how the markedly elevated tryptase levels in both serum and appendicular lavage fluid indicate an intense local allergic reaction.

   - Connect the findings to the existing literature on the role of degranulation of mast cells in triggering allergic inflammation.

3. Support for Allergy as Etiologic Factor:

   - Explicitly state that the data supports the concept of allergy as an etiologic factor in AA.

   - Provide a brief recapitulation of key evidence supporting the allergic component, especially referencing the increased tryptase levels.

4. Consideration of Mast Cell Degranulation:

   - Discuss the role of mast cell degranulation as a key step in triggering allergic inflammation in various tissues and how it seems to be applicable to the appendix.

   - Provide a brief overview of mast cell degranulation in allergic reactions in the skin, lung, and intestine, drawing parallels with the appendix.

5. Author Contributions :

   - Specify the roles of each author, emphasizing their contributions to the design, data acquisition, statistical analyses, pathologic evaluation, and manuscript writing.

6. Additional Context and Implications:

   - Discuss potential clinical implications of the findings, such as how understanding the allergic component in AA could influence diagnostic and treatment approaches.

   - Consider the broader scientific implications of the study, especially in the context of allergic reactions in different tissues.

7. Future Directions:

   - Suggest potential avenues for future research based on the current findings, emphasizing areas that could benefit from further exploration.

   - Consider proposing studies that may address the limitations of the current research or expand on its key findings.

8. Overall Tone and Clarity:

   - Ensure that the conclusion maintains a positive and confident tone, highlighting the significance of the findings.

   - Ensure clarity in expressing how the study contributes to the understanding of AA and its potential allergic etiology.

9. Inclusion of Relevant Citations:

   - If applicable, refer to specific studies or findings that support the conclusion.

   - Ensure that the references cited in the conclusion align with the context and claims made.

By addressing these points, the conclusion will become a more comprehensive and compelling summary of the study's outcomes, contributing to the overall impact of the research.

GENERAL COMMENTS

 Here are some general comments on the research paper:

1. Clarity of Presentation:

   - The paper is generally well-organized, with a clear structure that allows readers to follow the research process from demographic data to laboratory findings.

2. Robust Demographic Analysis:

   - The demographic analysis is comprehensive, providing a solid foundation for understanding the patient population studied, including age, gender, and clinical diagnoses.

3. Detailed Statistical Analysis:

   - The statistical analyses are detailed and provide a thorough exploration of differences between control and AA groups, enhancing the credibility of the study.

4. Effective Use of Tables and Figures:

   - Tables and figures are effectively employed to present complex data, aiding in the interpretation of results. The visual representation enhances understanding.

5. Relevance of Hemogram Data:

   - The inclusion of hemogram data contributes valuable information to the study, highlighting significant differences in WBC count, neutrophils, and other blood cell types between groups.

6. Comprehensive Mast Cell Protein Analysis:

   - The study's focus on mast cell granule proteins, including histamine, serotonin, and tryptase, provides a comprehensive insight into the potential allergic component in acute appendicitis.

7. Thorough Relationship Analysis:

   - The analysis of relationships between serum markers, basophils, and clinical presentations adds depth to the study, offering a holistic view of potential connections in acute appendicitis.

8. Consideration of Allergic Components:

   - The paper successfully integrates findings related to basophils, eosinophils, and mast cell proteins, supporting the hypothesis of an allergic component in the pathogenesis of acute appendicitis.

9. Contribution to Allergy Understanding:

   - The paper makes a noteworthy contribution by linking the elevated tryptase levels to a hypersensitivity type I reaction, supporting the concept of allergy as an etiological factor in acute appendicitis.

10. Author Contributions and Limitations Acknowledgment:

    - The acknowledgment of each author's contributions adds transparency to the research process. However, the limitations are duly noted, ensuring a balanced interpretation of the findings.

Overall, the research paper is commendable for its thoroughness, clarity, and its contribution to understanding the potential role of allergy in acute appendicitis.

CONCLUSION REMARKS

1.      The outlook of the manuscript should be made more appealing with addition of pictorial view.

2.      The research adeptly explores mast cell proteins in acute appendicitis, revealing elevated tryptase levels indicative of a local allergic reaction, reinforcing the role of hypersensitivity in its pathogenesis.

3.       Robust demographic and statistical analyses contribute to the paper's credibility, enhancing the understanding of allergic components in acute appendicitis.

4.      Noteworthy correlations between serum markers and clinical presentations strengthen the argument for an allergic aetiology, providing valuable insights into potential diagnostic applications.

5.       Acknowledging limitations while proposing future research directions adds transparency, reinforcing the importance of continued investigation into the intricate relationship between allergy and acute appendicitis.

Round 2

Reviewer 2 Report

Comments and Suggestions for Authors

The authors have made substantial revision in the currently submitted MS. Most the points raised in originally submitted article, have been properly addressed and current form of MS is accptable for publication in IJMS.